# Chordoma—Current Understanding and Modern Treatment Paradigms

**DOI:** 10.3390/jcm10051054

**Published:** 2021-03-04

**Authors:** Sean M. Barber, Saeed S. Sadrameli, Jonathan J. Lee, Jared S. Fridley, Bin S. Teh, Adetokunbo A. Oyelese, Albert E. Telfeian, Ziya L. Gokaslan

**Affiliations:** 1Department of Neurosurgery, Houston Methodist Neurological Institute, Houston Methodist Hospital, Houston, TX 77030, USA; sbarber@houstonmethodist.org (S.M.B.); ssadrameli@houstonmethodist.org (S.S.S.); jjlee@houstonmethodist.org (J.J.L.); 2Department of Neurosurgery, Rhode Island Hospital, The Warren Alpert Medical School at Brown University, Providence, RI 02903, USA; Jared.Fridley@Lifespan.org (J.S.F.); AOyelese@Lifespan.org (A.A.O.); ATelfeian@lifespan.org (A.E.T.); 3Department of Radiation Oncology, Houston Methodist Neurological Institute, Houston Methodist Hospital, Houston, TX 77030, USA; Bteh@houstonmethodist.org

**Keywords:** chordoma, spinal oncology, spinal tumor

## Abstract

Chordoma is a low-grade notochordal tumor of the skull base, mobile spine and sacrum which behaves malignantly and confers a poor prognosis despite indolent growth patterns. These tumors often present late in the disease course, tend to encapsulate adjacent neurovascular anatomy, seed resection cavities, recur locally and respond poorly to radiotherapy and conventional chemotherapy, all of which make chordomas challenging to treat. Extent of surgical resection and adequacy of surgical margins are the most important prognostic factors and thus patients with chordoma should be cared for by a highly experienced, multi-disciplinary surgical team in a quaternary center. Ongoing research into the molecular pathophysiology of chordoma has led to the discovery of several pathways that may serve as potential targets for molecular therapy, including a multitude of receptor tyrosine kinases (e.g., platelet-derived growth factor receptor [PDGFR], epidermal growth factor receptor [EGFR]), downstream cascades (e.g., phosphoinositide 3-kinase [PI3K]/protein kinase B [Akt]/mechanistic target of rapamycin [mTOR]), brachyury—a transcription factor expressed ubiquitously in chordoma but not in other tissues—and the fibroblast growth factor [FGF]/mitogen-activated protein kinase kinase [MEK]/extracellular signal-regulated kinase [ERK] pathway. In this review article, the pathophysiology, diagnosis and modern treatment paradigms of chordoma will be discussed with an emphasis on the ongoing research and advances in the field that may lead to improved outcomes for patients with this challenging disease.

## 1. Introduction

Chordoma is a notochordal tumor of the skull base, mobile spine and sacrum which remains a considerable treatment challenge despite over 150 years of surgical and chemoradiotherapeutic advances since the first description of the entity by Virchow in 1857 [1]. While histologically considered low-to-intermediate grade, these tumors behave in a malignant manner [2], inevitably recur despite aggressive therapy and confer considerable morbidity and mortality [3,4,5,6,7]. Tumor burden at the time of diagnosis is often large, margination is poor, tumor cells are relatively resistant to radiotherapy and chemotherapy [8,9,10] and there is a propensity for local growth along and around adjacent neurovascular anatomy. Together these factors complicate the ongoing search for a definitive “cure” for chordoma.

Extent of tumor resection is the most crucial factor in the prognosis of chordoma, with wide, en bloc resection being the surgical gold standard. Treatment by an experienced surgeon in coordination with a multidisciplinary team is paramount to achieving the best patient outcomes [5,6,11,12,13,14,15,16,17,18]. Though chordoma has been traditionally considered resistant to conventional radiotherapy and chemotherapy, advances in radiation targeting and treatment schemes have improved our ability to deliver high doses of radiation to the tumor itself while minimizing radiation toxicity to surrounding structures [19,20,21]. Promising research into the underlying genetic and molecular pathophysiology of chordoma has also led to advanced, targeted chemotherapeutic agents [22,23,24].

## 2. Embryology and Pathophysiology

### 2.1. Embryology of the Notochord

Chordomas are presumed to derive from undifferentiated, extradural, vestigial remnants of the notochord, an embryonic structure that coordinates cell fate and development. This presumption is based on studies showing that the sites at which chordomas originate correspond well to the location of embryological notochord cell rests [25,26], as well as the fact that brachyury—a transcription factor required for notochord development and expressed within undifferentiated embryonic notochord—is overexpressed in chordomas [27,28].

The notochord itself is a longitudinal, axial structure present centrally (with respect to dorsal/ventral and left/right axes) within humans and all other members of the phylum Chordata during embryological development. It courses cranially from the sacrum within the boundaries of what will become the vertebral bodies, exiting this path only briefly—ventral to what will become the clivus—to contact pharyngeal endoderm before returning dorsally, where it terminates at the level of the dorsum sellae of the sphenoid bone. The notochord provides position and fate information to the developing embryo and serves a structural role, acting as a primitive axial skeleton during embryonic development. In some vertebrates, the notochord is present throughout life and functions as a support structure during locomotion and other tasks. In humans and other higher vertebrates, however, the notochord ossifies to form the vertebral bodies and completely regresses within the first few years of life. Following this process only a small remnant of the notochord remaining as the nucleus pulposis of intervertebral discs [11,26,29].

Chordomas do not, however, arise from the nucleus pulposis. In fact, chordomas seldom involve the intervertebral disc—but rather arise from aberrant vestiges of the notochord that failed to properly regress. Multiple notochordal vestiges have been described from which chordoma may arise, such as ecchordosis physalifora, a hamartomatous mass found dorsal to the clivus, which, according to a retrospective series by Mehnert et al., may be seen incidentally on magnetic resonance imaging (MRI) in 1.7% of patients [30]. Other examples of vestigial notochordal entities include benign notochord cell tumor (BNCT) and parachordoma. BNCTs may be found within the clivus, sacrum or mobile spine. The fact that BNCTs are known to have a potential for malignant transformation and that the anatomical locations in which BNCTs occur overlap with that of chordoma, suggests that BNCTs—and perhaps other notochordal vestigial remnants—are the prototypical entities from which chordoma originates [31].

Though BNCT and other ectopic notochordal remnants have been hypothesized to serve as the precursors for chordoma [31], the incidence of vestigial notochordal remnants such as BNCT is much higher than that of chordoma and thus, the large majority of these notochordal remnants are presumed to lie dormant indefinitely, unless spurred by some stimulus—whether exogenous or endogenous—to mutate and become malignant [32].

### 2.2. Chordomagenesis

Chordomas may theoretically arise from any anatomical location along the length of the notochord’s former embryological course. Though chordomas were previously thought to have a predilection for the clivus and sacrum, recent epidemiological studies suggest that they occur with a similar incidence in the skull base, mobile spine and sacrum [3]. The precise mechanism underlying the transformation from notochordal vestige to chordoma is not well-understood, although recent studies have highlighted several chromosomal and cell cycle aberrations thought to contribute to chordomagenesis. Overexpression of both p53 and CDK4, for instance, which function in the G1 phase of the cell cycle, has been shown to be present in some chordomas and is correlated with decreased overall survival [33]. Loss of 1p36 and loss of heterozygosity at 9p also correlate with more aggressive chordoma behavior and decreased overall survival [34].

The T gene (*6p27*) encodes brachyury, which, as mentioned previously, is a transcription factor required for notochord development from mesodermal elements. Brachyury is expressed transiently in embryonic notochord and normally silenced in post-developmental tissues but has been shown to be aberrantly re-expressed in chordoma and some evidence exists for its causative role in chordomagenesis. Knockdown of brachyury expression, for instance, leads to suppression of chordoma cell line growth in vitro [35]. Studies examining familial chordoma cohorts identified a recurrent germ-line duplication in *6p27* [24,28], which contains the brachyury gene, although this finding is only present in a small percentage of patients with sporadic chordomas [36]. Brachyury expression and T gene copy number gains have also been found to correlate with progression-free survival (PFS) in skull-base chordomas [37,38] but not spinal chordomas [39].

The fibroblast growth factor (FGF)/mitogen-activated protein kinase kinase (MEK)/extracellular signal-regulated kinase (ERK) pathway mediates expression and signaling of brachyury in chordomas and may also play a role in chordomagenesis. Fibroblast growth factor is a known regulator of brachyury expression in normal tissue and FGF2, FGF3, MEK and ERK have all been shown to be expressed in cultured chordoma cell lines. Exposing these cell lines to an FGFR inhibitor reduced MEK/ERK phosphorylation and brachyury expression, inducing apoptosis and restricting cell growth [40]. Furthermore, small hairpin RNA knockdown of brachyury blocked the effect of FGF2 on epithelial-mesenchymal transition, a process thought to be involved in carcinoma progression and metastasis [40].

Receptor tyrosine kinases (RTKs) are another area of interest in the pathogenesis of chordoma and other cancers. RTKs are transmembrane proteins that translate extracellular stimuli into intracellular signaling cascades responsible for cell growth, differentiation and proliferation. Dysfunctional signaling within an RTK cascade can thus lead to the aberrant behavior observed in tumor cells and oncogenic mutations have been documented in a number of different RTK families (Figure 1). With chordoma in particular, platelet-derived growth factor (PDGF) receptor (PDGFR), epidermal growth factor (EGF) receptor (EGFR) and hepatocyte growth factor (HGF) receptor (c-Met) are three RTKs thought to play a role in pathogenesis and malignant potential, as each has been shown to be overexpressed in chordoma [41,42]. Furthermore, the phosphoinositide 3-kinase (PI3K)/protein kinase B (Akt)/mammalian target of rapamycin (mTOR) pathway is downstream of PDGFR, EGFR and c-Met and has been shown to be hyperactive in chordomas [43], suggesting that this pathway may be a unifying mechanism through which abnormal signaling by a variety of different RTKs underlies the pathophysiology of chordoma.

## 3. Epidemiology

The incidence of chordoma is only 0.08 in 100,000 individuals [3]. Males (incidence rate 0.10) are more commonly affected than females (incidence rate 0.06), with a peak age between 50–60 years (median, 58.5 years). Children and adolescents are rarely affected (<5% of cases) [3].

Chordomas may present in any location along the axis of what was formerly the notochord and accordingly, population studies have shown that they arise with a similar distribution in the sacrum (29.2%), mobile spine (32.8%) and skull base (32%) [3]. Female patients and those presenting at a younger age (<26 years) have a higher incidence of skull base chordomas. Chordomas are the most common primary sacral tumor (>50% of cases) [44,45] and account for 17% of primary tumors of the mobile spine but only 1.4% of all primary bony malignancies and 0.2% of skull base tumors [11]. Within the mobile spine, chordomas occur with greatest frequency in the high cervical spine (C2 and C3) [11,46] and within the sacrum, they most commonly involve the 4th and 5th sacral vertebrae [47]. Occasionally, chordomas have been described in paramedian locations (e.g., petrous apex, jugular foramen, Meckel’s cave), presumably due to aberrant or variant “forking” of the notochordal course during development [15,48]. Chordomas may also arise within the nasopharyngeal soft tissues without bony involvement, due to the normal embryological “kink” of the notochord as it briefly exits the clivus ventrally to contact pharyngeal endoderm [25,49]. While chordomas usually occur in adults, rare cases of skull base and spinal chordomas in the pediatric population have been reported. These tumors are often poorly differentiated with high risk of metastasis and are associated with loss of SMARCB1/INI1 protein [50,51]. The genetic association in these pediatric or de-differentiated chordomas has opened therapeutic avenues for potential targeting with EZH2 inhibitors (e.g., Tazemetostat) [50,52].

Chordomas are typically sporadic but a number of suspected familial forms have been documented, with a variety of familial genetic mutations identified in these cases, including recurrent germ-line mutations in the T gene (*6p27*), which encodes brachyury, as mentioned previously [28]. Chordomas have also been reported to arise in association with other pathological syndromes, such as tuberous sclerosis complex, Ollier disease and Maffuci syndrome [53,54,55].

## 4. Clinical Presentation

The clinical symptoms with which chordomas present will vary by location. Patients with skull base chordomas often present with headaches, cranial neuropathies and endocrinopathies, whereas patients with chordomas of the mobile spine and sacrum may present with localized pain, radiculopathies, myelopathy and/or bowel/bladder dysfunction. Given their indolent growth patterns, many chordomas are relatively large at presentation and may even be discovered as a palpable mass. Chordomas of the cervical spine may invade into the cervical soft tissues, leading to dysphagia and airway obstruction and chordomas of the sacrum may grow into the pre-sacral space and pelvis, causing constipation, urinary retention and visceral pain. Most chordomas originate in the extradural space, although some violate the dura and spread intradurally and several cases of isolated intradural and intraparenchymal chordomas have been reported.

## 5. Diagnosis

### 5.1. Imaging

On computed tomography (CT), chordomas appear as a midline, well-circumscribed and expansile soft-tissue mass with lytic destruction of surrounding bone (Figure 2). Intratumoral hyperdensities may be seen and are thought to represent sequestrate of normal bone—except in chondroid chordoma variants, in which case true intratumoral calcifications are found. The tumor mass typically enhances with contrast [56,57]. Low-density areas may be seen within the soft-tissue mass on CT, correlating with myxoid, gelatinous portions of the tumor demonstrated during gross examination. Within the mobile spine and sacrum, chordoma typically involves one or more vertebral bodies with or without a paraspinal/epidural component but the intervertebral discs are often spared.

On T1-weighted magnetic resonance images (MRI), chordomas exhibit intermediate to low-signal intensity (Figure 2). Small foci of T1 hyperintensity may be seen as a result of intratumoral microhemorrhage (which will be dark on susceptibility-weighted/gradient echo sequences) and mucus pooling. Chordoma is frequently hyperintense on T2-weighted imaging with hypointense septations and small foci of hypointensity corresponding to hemorrhage, calcification and mucus pooling [56,57]. Most chordomas will enhance moderately-to-intensely with gadolinium in a variable “honeycomb” pattern and some studies suggest that the degree of contrast enhancement correlates with aggressive tumor behavior and risk of recurrence/progression after surgical resection [58].

Conventional angiographic evaluation of chordomas is typically non-specific and abnormal vasculature is rare. Although intracranial chordomas and those originating in the cervical spine have a propensity to envelope major arteries, luminal narrowing is not frequently seen due to the soft consistency of most chordomas. MR angiography and venography may be useful when evaluating for encasement of important vessels but conventional angiography is reserved for cases in which significant encasement or displacement of major arteries is suspected or when vessel test occlusion and/or embolization is being considered in efforts to achieve a more radical surgical resection [56].

### 5.2. Histopathology

Though the diagnosis of chordoma may be suspected based on patient presentation and radiographic imaging, diagnostic certainty requires histopathological evaluation. CT-guided needle biopsy and tissue diagnosis is often recommended prior to definitive surgical intervention, as the differential diagnosis for chordomas includes other entities for which treatment approaches may vary [45].

Virchow was the first to characterize the histology of chordoma [1]. He described it as a “physaliphorous” (i.e., vacuolated or bubble-bearing) mass, in reference to the numerous cytoplasmic vacuoles by which it is characterized to this day [2]. Chordoma is composed of numerous discrete lobules separated by fibrous bands. Tumor cells display abundant, pale, cytoplasm and are arranged in sheets, cords or as solitary cells floating in a myxoid stroma (Figure 3). Nuclear atypia is only mild to moderate and mitoses are infrequent. Immunohistochemistry studies will reveal a reaction with antibodies against S100 protein, pan-keratin, low molecular cytokeratins and epithelial membrane antigen (EMA) [2].

Four subtypes of chordoma are currently recognized: conventional (which is the most common), chondroid (in which areas mimicking hyaline or myxoid cartilage are seen), poorly differentiated and dedifferentiated or sarcomatoid chordoma (chordoma associated with a high-grade sarcoma, which accounts for only 5% of cases and confers the worst prognosis) [2,58,59,60]. Because chondroid chordomas share pathological features with chondrosarcoma and because their clinical presentation and radiographic appearance are often similar, distinguishing between the two histologically may be challenging. Some have suggested that brachyury, the notochordal transcription factor, may be useful in making the distinction, as it is expressed by the large majority of chordomas but not chondrosarcomas [27,61].

## 6. Treatment

Chordomas pose a considerable treatment challenge due to their midline location, predilection for involving critical neurovascular anatomy, indolent growth patterns, tendency to seed/recur and resistance to traditional chemoradiotherapeutic modalities. Advances in surgical techniques and approaches, image guidance, radiotherapy strategies and the emergence of promising targeted molecular therapies, however, are beginning to change the outlook of the disease. In order to achieve the best possible patient outcomes, it is crucial that patients with chordomas are treated by multidisciplinary teams in experienced quaternary centers, where a well-planned biopsy, appropriate staging, maximal safe surgical resection, molecular sequencing, modern medical therapy, radiotherapy and relevant clinical trials are all accessible and included in the decision-making process. Performance of even some invasive diagnostic procedures outside of an experienced center has been shown to negatively impact patient prognosis [12].

### 6.1. Surgery

Surgical resection is currently the mainstay of treatment for chordoma and extent of surgical resection is one of the most important prognostic factors for patients with this disease [4,6,13,46,60,62,63,64,65,66,67]. Chordomas are known to have a propensity for seeding tumor cells throughout a surgical corridor, contributing to tumor recurrence [68,69] and thus en bloc resection without capsule violation—when feasible—has often been considered the surgical gold standard. Chordomas occur most commonly in the midline, however, and tend to invade in and around critical neurovascular structures, making en bloc resection a challenge without imparting considerable morbidity. Furthermore, the concept of “en bloc resection,” in the absence of a description of margins by an experienced pathologist, is imprecise and does not adequately distinguish between margins that contain neoplastic tissue and those that do not. Decision-making regarding surgical approach and extent of resection must be undertaken as a cost-benefit analysis with consideration of a multitude of patient- and tumor-specific factors, including tumor location, the neurovascular anatomy involved and patient functional status, among others, with the goal being to achieve as radical of resection as possible at first presentation while avoiding morbidity.

#### 6.1.1. Biopsy and Preoperative Workup

Biopsy of suspected chordomas should be performed in an experienced center prior to surgical intervention, as the diagnosis may significantly alter the overall plan. Recommendations for surgical margins may vary based on the grade of tumor and the presence of metastases and in some cases, radiotherapy and/or palliative procedures may be recommended rather than aggressive resection. CT-guided needle biopsy is recommended rather than open biopsy for suspected chordomas of the mobile spine and sacrum, as seeding is likely to occur as a result of open biopsy, but when a needle is used, the tract may be resected at the time of definitive tumor resection.

#### 6.1.2. Skull Base Chordomas

The preferred surgical approach for skull base chordomas and chordomas of the craniocervical junction depends on the precise location and extent of the tumor, with anterior, anterolateral and lateral approaches being favored. Often a combination of approaches is used and in many cases, several repeat resections are needed throughout the course of the disease. In a large series by Wang et al., 238 patients with skull base chordomas treated over a period of 10 years were reviewed and the tumor location, surgical approach and outcomes were described [4].

Conventional and expanded endoscopic endonasal approaches (EEA) are favored by many surgeons for resection of chordomas and other skull base tumors due to the minimally invasive nature of the approach (a physiologic approach corridor which obviates the need for a skin incision) and the increasingly favorable outcomes reported in the literature with this technique [70,71]. Chordomas, in particular, are often situated ventrally and are thus amenable to an EEA. Every approach has advantages and disadvantages, however and the decision to use a given approach—or multiple approaches—must take a variety of patient and tumor-specific factors into account as well as the overall surgical goal in each case.

#### 6.1.3. Mobile Spine Chordomas

Given the tendency for chordoma to seed and recur locally as a result of capsule violation, precise characterization of resection margins is paramount when discussing surgical approach and outcomes and the term “en bloc” is lacking in this regard. As such, the Enneking or the Weinstain-Boriani-Biagini classifications are the preferred means of characterizing spinal and sacral chordomas and providing recommendations regarding margins of resection [6,72,73,74,75]. The Enneking classification was originally developed for use with musculoskeletal tumors and several modifications must be made in order to account for the dura and intradural anatomy unique to the spine, but the essence of the Enneking classification in chordomas is unchanged: the tumor should ideally not be entered during resection. When a margin of normal tissue is present around the tumor specimen the resection is considered “wide” and when a margin of pseudocapsule (i.e., without neoplastic cells) is present around the tumor specimen the resection is considered “marginal” (in chordomas encroaching on the epidural space, a wide resection margin is not theoretically possible without resection of dura and/or neural elements). When the tumor is entered during the resection, this is termed an “intralesional” resection.

#### 6.1.4. Sacral Chordomas

As with chordomas of the mobile spine, sacral chordomas are suited to classification according to the Enneking staging system and the recommendations for intralesional, marginal or wide resections margins translate well to sacral chordomas. Performing wide or marginal resections of sacral chordomas, however, is made challenging by the uniquely complicated surrounding anatomy, including nerves contributing to lower extremity function, sexual function and bladder/bowel function, bony attachments to the lumbar spine and pelvis, the iliac vessels, surrounding gluteal and piriformis musculature, as well as the retroperitoneal and pelvic viscera, which may be involved depending on the extent of the tumor. Multiple surgical specialties (e.g., neurosurgery, orthopedic surgery, vascular surgery, general surgery, plastic surgery) will likely need to be involved and multiple approaches (i.e., anterior, lateral, posterior) may be required to achieve the desired margins without capsule violation.

Total sacrectomy is frequently performed in two stages, with an anterior approach being utilized first to create the desired margins between normal anatomy and the anterior aspect of the tumor, ligate and mobilize vessels and nerve roots as needed and create partial anterior sacroiliac osteotomies. A rectus abdominis myocutaneous flap with vascular supply from the inferior epigastric vessels may also be harvested prior to abdominal closure and stowed within the abdominal cavity to be pulled-through for coverage and closure after the second, posterior stage of the surgery, during which the sacrum and involved tumor is removed. Some authors have demonstrated favorable outcomes with posterior-only approaches for en bloc sacrectomy [76], although others have suggested that the posterior-only approach is most suitable for lesions located at S3 and below [14]..

In many cases, in order to achieve the desired margins, nerve roots will need to be sacrificed and the functional results of such a sacrifice will vary depending on the roots involved. Low sacral amputations often lead to sacrifice of the roots distal to S3, which tends to result in minimal deficit, with the exception of a variable reduction in perineal sensation and sexual function—though sphincter function is typically preserved [77]. Mid-sacral amputations in which one or more of the S2 and S3 roots are removed may lead to saddle anesthesia and sphincter dysfunction. Preservation of at least one S3 root has been reported to preserve functional continence in some cases [77,78]. High sacral amputations and total sacrectomies (in which S1 roots are removed) lead to expected deficits in plantar flexion as well as loss of sphincter control and sexual function, although, as mentioned previously, sphincter control may be partially preserved with only unilateral sacral root resection [77] but this is variable [79].

High sacral amputation often leads to impaired stability and advanced instrumentation techniques described in detail elsewhere may be required to reconstruct the pelvic ring and re-establish spinosacral and sacropelvic stability in order for patients to be safely mobilized [78,80,81].

Wound dehiscence, wound infections and CSF leak are the most commonly cited complication of sacrectomy for chordoma, with as many as 1 in 4 patients requiring further surgery as a result [13,14,82]. Posterior sacral incisions are often in close proximity to the anus and wound contamination is thus a concern. Plastic surgery involvement, the use of myocutaneous flaps for closure and even prophylactic diverting colostomy may be required to prevent wound related complications [13,14,71,76]. Other reported perioperative complications include deep venous thrombosis, pulmonary emboli, pneumonia, myocardial infarction, inadvertent bowel perforation, pelvic fatigue fractures, hemorrhage, CSF leak, muscle necrosis, neurological deficits in motor, sensory or sphincter control and even death [8,13,14,82].

As with chordomas in the skull base and mobile spine, the adequacy of resection margins has the greatest influence on prognosis, overall survival and risk of local recurrence [12,13,14,67,79], yet wide or marginal resections of sacral chordoma are achieved in only 40 – 55.6% of cases [8,13,14]. Even in cases of aggressive resection, recurrence is an unfortunate inevitability and overall survival is relatively poor, especially considering the relatively low/intermediate-grade nature of these tumors [3,5,8,12,13,14,16,45,67]. After local recurrence, the decision to proceed with repeat resection, adjuvant therapy and/or palliation will depend on the degree of recurrence, the presence/absence of systemic disease and patient-specific factors such as functional status [7].

### 6.2. Radiotherapy

By most accounts, conventional radiotherapy is ineffective as a stand-alone treatment or when coupled with intralesional resections for chordoma [9,46]. The effectiveness of even modern radiotherapy in chordoma is a matter of controversy, with several recent series demonstrating little benefit [17,83,84]. Emerging evidence suggests, however, that modern radiotherapy may have a role as an adjuvant to aggressive resections; particularly when performed early in the disease course [9,15,21,85,86] and some evidence exists that high-dose stereotactic radiotherapy provides durable local control even when used as a definitive treatment for patients unable to undergo an aggressive surgical resection [87].

One of the challenges with regard to radiotherapy for chordomas lies in the crucial—and relatively radiosensitive—neurovascular anatomy with which chordomas are often intimately involved. Image-guidance, stereotaxis, dosimetric planning and the use of hadrons (e.g., protons and carbon ions) have improved upon our ability to deliver high doses of ionizing radiation to chordomas while sparing nearby anatomy. Hadron radiotherapy takes advantage of the Bragg peak effect, wherein the maximum dose of radiation delivered by heavy, charged particles like protons and carbon ions occurs at a precise depth (i.e., the Bragg peak), immediately before the particle comes to rest, with relatively small doses of radiation delivered throughout the rest of a particle’s course [88,89]. The depth and width of the Bragg peak varies with beam energy and the composition of tissues in the beam path, both of which can be measured and controlled. This allows for delivery of high radiation doses to target tissues while sparing skin and other anatomy nearby. Photons, in contrast, reach peak energy deposition within less than a centimeter of tissue penetration, thereafter depositing energy at an exponentially decreasing rate with increasing depth in tissue. Thus, concentrating photon radiotherapy dose within a deep-seated lesion with sparing of surrounding anatomy can only be accomplished using multiple beam directions, resulting in a greater net energy deposition within the patient when compared to proton therapy [90].

Although proton beam therapy is the radiotherapy delivery method preferred by some quaternary centers for the treatment of chordoma, the stereotactic delivery of photon therapy has been shown to confer similar durability of local control with an acceptable toxicity profile [87] and no evidence currently exists to conclusively demonstrate the superiority of one radiotherapy modality over another [84]. Perhaps more important than the type of particle delivered is the timing of radiation with regards to surgery [84,85], as several studies have demonstrated that rates of local control are improved when radiotherapy is delivered at the time of primary resection, rather than when the tumor recurs [91]. Others have demonstrated favorable outcomes with pre-operative radiotherapy followed by a post-operative boost [21].

Brachytherapy is another means by which radiation may be delivered to chordomas. Dural plaques (e.g., phosphorus-32 [P32], yttrium) have been used by some authors as an adjuvant to external beam radiotherapy to improve dose delivery to dural margins with less risk of toxicity to the spinal cord in tumors with epidural extension [92,93]. Objective evidence of superiority to external beam radiotherapy alone in chordomas is lacking, however.

Finally, carbon ion radiotherapy has been described in the recent literature as an effective treatment option for skull base chordoma with acceptable late toxicity and local control [94]. Further research is warranted to evaluate long term efficacy and side effects of carbon ion therapy.

### 6.3. Medical Treatments

Chordoma is notoriously insensitive to traditional chemotherapeutic agents [93,95] but ongoing research into the molecular pathways underlying chordoma pathophysiology have led to a number of promising targeted molecular agents. Imatinib and sunitinib, for example, are both tyrosine kinase inhibitors with activity against PDGFR, KIT receptors, vascular endothelial growth factor receptors and other molecular pathway elements known to be overexpressed in chordoma. Both have shown modest efficacy in early clinical trials, although further study is needed [23,96]. Overexpression of EGFR and c-MET in some chordomas has led to the use of several agents with anti-EGFR activity, including erlotinib and lapatinib, both of which have been suggested to have some degree of clinical efficacy by small series and/or case reports [97,98]. Afatinib, another EGFR inhibitor, demonstrated anti-proliferative activity against a number of chordoma cell lines in vivo [99] and is currently involved in a Phase II clinical trial for patients with advanced or metastatic chordoma (NCT03083678, www.clinicaltrials.gov accessed on 31 January 2021).

Cyclin D-dependent kinases (CDK4 and CDK6) are another target for chordoma molecular therapy. These oncoproteins regulate cell cycle progression through the G1 phase into the S phase and have been shown to be overexpressed in chordoma cell lines and tissue samples due to loss of the *CDKN2A* gene and its protein product, p16INK4a [100]. Palbociclib, a CDK4/6 inhibitor, has been demonstrated to have in vitro [99,100] efficacy against chordoma cells and is part of a Phase II trial for advanced chordoma and other tumors (NCT03110744, www.clinicaltrials.gov accessed on 31 January 2021) (Figure 4).

Immune checkpoint inhibitors are a potential line of therapy in chordomas and other cancers, as immune checkpoints have been demonstrated to impair the tumor-killing ability of T-lymphocytes. Anti-programmed cell death protein-1 (PD-1) and anti-programmed cell death ligand-1 (PD-L1) are immune checkpoints shown to be expressed in chordoma cell lines [101,102]. Avelumab (anti-PD-L1 monoclonal antibody) has been shown to have efficacy against PD-L1-expressing chordoma cells in vitro [103] and some anecdotal evidence exists of nivolumab’s (anti-PD-1 monoclonal antibody) in vivo efficacy [104]. Nivolumab is currently part of a phase II trial for patients with chordoma and other rare CNS tumors (NCT03173950, www.clinicaltrials.gov accessed on 31 January 2021), as well as a phase I trial studying the safety and initial effectiveness of the antibody with or without stereotactic radiosurgery for chordoma and other advanced cancers (NCT02989636, www.clinicaltrials.gov accessed on 31 January 2021).

Brachyury is another potential target for molecular therapy, as the protein is present semi-ubiquitously in chordomas and brachyury silencing has been shown to cause growth arrest and senescence in chordoma cell lines in vitro [35]. Brachyury has also been shown to promote epithelial-mesenchymal transition (EMT), a process thought to be involved in tumor progression and metastasis, through which cancer cells transform from an epithelial phenotype into a motile, mesenchymal one [105]. Though no direct inhibitors of brachyury currently exist for clinical use, a number of vaccine platforms have been developed to induce immunization against brachyury [106]. A heat-killed recombinant yeast expressing brachyury (GI-6301) was shown to elicit brachyury-specific T-cell responses that reduce tumor burden in a mouse model of lung metastases [107] and is currently involved in a Phase II human clinical trial for patients with chordoma (NCT02383498; www.clinicaltrials.gov accessed on 31 January 2021). A modified vaccinia Ankara (MVA) poxviral vaccine vector has also been developed that encodes human brachyury as well as a triad of costimulatory molecules (B7-1, ICAM-1 and LFA-3) and is part of a Phase I clinical trial in patients with advanced cancers (NCT02179515; www.clinicaltrials.gov accessed on 31 January 2021). Finally, an adenovirus serotype 5 (Ad5) vaccine encoding brachyury, carcinoembryonic antigen (CEA) and mucin-1, cell surface associated (MUC1) has been developed and is currently being tested in a Phase I clinical trial for patients with advanced cancers (NCT03384316, www.clinicaltrials.gov accessed on 31 January 2021). Finally, several studies have discussed pharmacological inhibition of the chordoma oncogene, *TBXT* (transcription factor brachyury), via inhibition of H3K27-demethylases as a promising novel therapy altering gene networks necessary for tumor survival [108,109,110].

## 7. Prognosis

Aggressive resection (i.e., wide or marginal when feasible) is the most important prognostic indicator for patients with chordoma [5,6,8,12,13,14,75]. Data from epidemiologic studies indicate that the overall mean survival for patients with chordoma is 6.29 years, with 67.6% surviving at 5 years, 39.9% at 10 years and 13.1% at 20 years [3]. These data were collected over 20 years ago, however and only 84% of these patients underwent surgery. Furthermore, the tumor location/extent and surgical margins achieved in these patients are unclear. With modern, aggressive surgery, 10-year overall survival rates of 95% have been reported for skull base chordomas [89] and 58–100% for chordomas of the mobile spine and sacrum [6,13,14,17].

Other reported prognostic factors for chordomas include pathology (with dedifferentiated chordomas having a poorer prognosis), history of prior resection (secondary patients referred due to recurrence after a previous resection have a poorer prognosis) and in sacral chordomas, the presence of muscle/sacroiliac joint involvement and higher location (above S3), all of which are reported to confer a poorer prognosis [12,58,59,78].

Even among conventional chordomas, however, biologically distinct tumor types appear to be present, with some chordomas behaving more aggressively than others despite apparently conventional histology [11]. Additionally, several studies into the underlying molecular pathophysiology of chordomas have suggested that skull base and spinal/sacral chordomas are in some ways biologically distinct [33]. Brachyury expression and T gene copy number gains have been found to correlate with progression-free survival (PFS) in skull-base chordomas [37,38] but not spinal chordomas, for example [39]. Although our current understanding of the impact of tumor molecular biology on prognosis is limited, the ability to identify more aggressive subtypes of chordoma after resection could be useful for decision-making regarding repeat resections and adjuvant chemoradiotherapy. Zenonos et al. prospectively evaluated 105 clival chordoma samples with fluorescence in situ hybridization (FISH) of chromosomal loci 1p36 and 9p21 as well as immunohistochemistry for Ki-67 in the interest of identifying molecular markers for prognosis, finding that 1p36 deletions and 9p21 homozygous deletions were predictive of poorer progression-free survival after surgery or radiotherapy, independent of Ki-67 [111].

## 8. Expert Opinion

Chordoma continues to present a considerable treatment challenge due to a number of factors, including: (1) indolent growth rate and resultant tendency to present late in the disease course, (2) propensity for invading through tissue planes, encapsulating critical nearby anatomy and seeding after resection and (3) its relative insensitivity to conventional radiotherapy and chemotherapy.

Extent of resection and surgical margins are currently the most important prognostic factors in chordoma and thus it is crucial that patients with chordoma are referred soon after presentation to an experienced quaternary care center, where every step of the diagnostic and therapeutic process can be controlled for the best patient outcomes. Biopsy and surgery should be carefully planned and executed by an experienced surgeon in association with a number of other surgical specialties so as to result in the widest margins and least likelihood of recurrence possible. Molecular sequencing, radiotherapy and clinical trials for promising molecular targeted agents should ideally all be available to patients and incorporated into the decision-making process.

The role of radiotherapy in the treatment of chordoma is controversial. Evidence regarding the benefit of radiotherapy is contradictory at times and little consensus exists for its role in chordoma treatment. Part of this discrepancy in findings, however, may be related to bias and confounding, in part due to differences in execution and terminology between studies. Currently, some centers proceed with adjuvant radiotherapy only in cases of incomplete and intralesional resections, which may subject the outcomes to selection bias. Furthermore, the terminology utilized in many series regarding the degree of resection is imprecise and this further confuses the outcomes. In spinal and sacral chordomas, in particular, we recommend that the term “en bloc” be avoided in lieu of Enneking-appropriateness, as the latter provides more consistent information regarding tumor violation during resection and has been shown to correlate well with local recurrence and overall survival.

Wide or marginal resections have the greatest impact on local recurrence and survival and should thus be pursued whenever possible but always with consideration of risk and in a manner that balances extent of resection with patient morbidity. Often our focus is primarily on overall survival and recurrence and yet, for many patients, quality of life is at least equal in value to quantity.

Though the prognosis of patients with chordoma is still relatively poor, our understanding of the genetic and molecular basis for chordoma pathophysiology continues to blossom and this is likely to hold the greatest promise in terms of providing a true cure. Early clinical outcomes of agents targeting PDGFR, EGFR, c-Met and other elements of the chordoma molecular pathophysiological cascade have been modest but our understanding continues to evolve. At the present time, surgical resection is the preferred treatment for chordoma but in time, advances in targeted medical therapy and radiotherapy delivery may lead to these modalities playing a more prominent role in the treatment of chordoma, with aggressive surgery being reserved only for the most advanced cases.

## Figures and Tables

**Figure 1 jcm-10-01054-f001:**
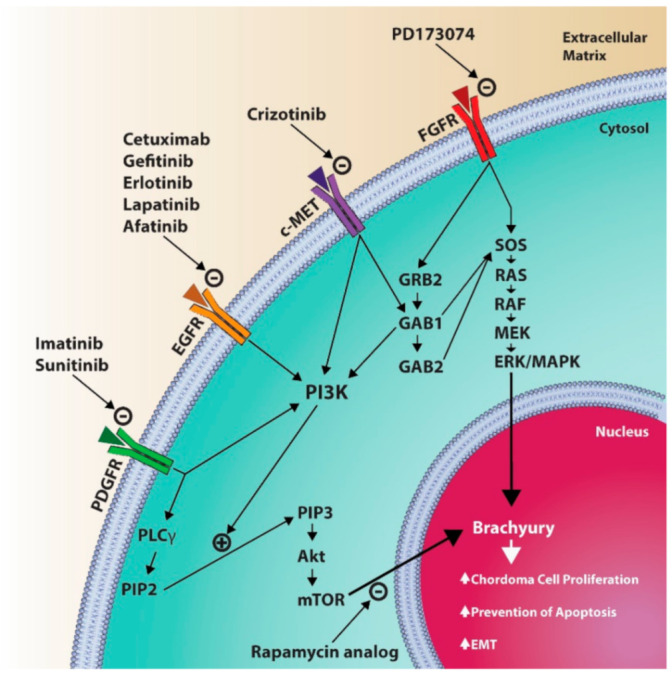
Diagram illustrating some of the molecular pathways involved in chordoma pathophysiology. PDGFR, platelet-derived growth factor receptor; EGFR, epidermal growth factor receptor; c-MET, hepatocyte growth factor receptor; FGFR, fibroblast growth factor receptor; PLCγ, phospholipase C gamma; PIP2, phosphatidylinositol 4,5 bisphosphate; PIP3, phosphatidylinositol 3,4,5 triphosphate; Akt, active human serine/threonine protein kinase; mTOR, mammalian target of rapamycin; PI3K, phosphoinositide 3 kinases; GRB2, growth factor receptor bound protein 2; GAB1, GRB-associated binding protein 1; GAB2, GRB-associated binding protein 2; SOS, son of sevenless protein (guanine nucleotide exchange factor); RAS, rat sarcoma protein; RAF, rapidly accelerated fibrosarcoma protein; MEK, mitogen activated protein kinase kinase; ERK/MAPK, extracellular signal-related kinases/mitogen activated protein kinases; EMT, epithelial-mesenchymal transition.

**Figure 2 jcm-10-01054-f002:**
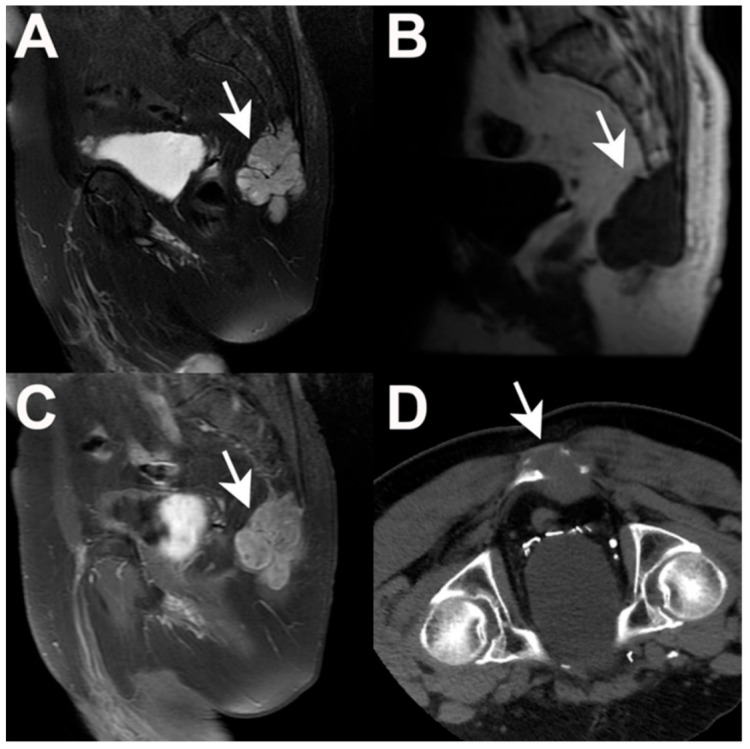
Radiographic appearance of a sacral chordoma. Sagittal T2-weighted MRI (**A**) demonstrating predominant hyperintensity with hypointense septations, scattered calcification and microhemorrhage. Sagittal T1 pre-contrast MRI (**B**) demonstrating diffuse hypointense signal. Sagittal T1-weighted post-contrast MRI (**C**) demonstrating marked, heterogenous contrast enhancement. Axial CT (**D**) demonstrating an expansile soft tissue mass with lytic bony destruction.

**Figure 3 jcm-10-01054-f003:**
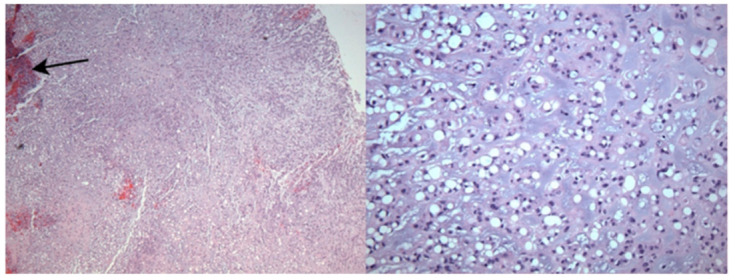
Histological findings in chordoma. (**Left**, ×40) Chords of tumor cells in a myxoid background with occasional microhemorrhage/calcification (arrow). (**Right**, ×400) Physaliferous cells with multiple intracytoplasmic vacuoles.

**Figure 4 jcm-10-01054-f004:**
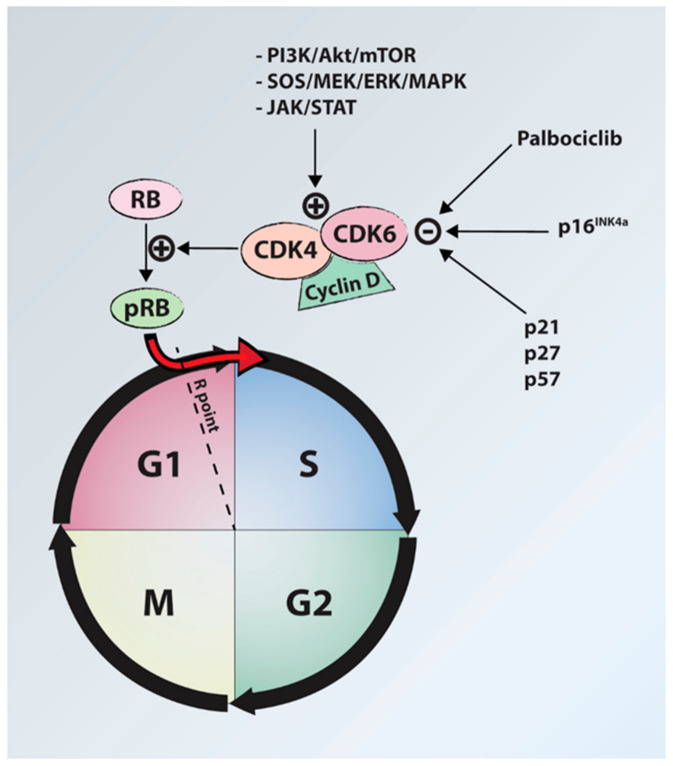
Diagram illustrating the role of cyclin D-dependent kinases (CDK4, CDK6) in progression from G1 phase to S phase. Palbociclib inhibits the action of CDK4/6, halting cell cycle progression. P16INK4a is often deficient in chordomas due to loss of the CDKN2A gene. RB, retinoblastoma protein; pRB, phosphorylated retinoblastoma protein; PI3K, phosphoinositide 3 kinases; Akt, active human serine/threonine protein kinase; mTOR, mammalian target of rapamycin; SOS, son of sevenless protein (guanine nucleotide exchange factor); MEK, mitogen activated protein kinase kinase; ERK/MAPK, extracellular signal-related kinases/mitogen activated protein kinases; JAK, janus kinase; STAT, signal transducer and activator of transcription.

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
