# Peer review of "Chordoma—Current Understanding and Modern Treatment Paradigms"

_jcm, 2021, doi:10.3390/jcm10051054_

Round 1

Reviewer 1 Report

The authors performed an exhaustive and informative narrative review on current concepts and knowledge on chordoma. I would advise only minor revisions:(Page 7, Line 239) The concept of a better prognosis in chondroid chordomas belongs to the past and is being changed in the last research studies, please add reference to support your sentence.

The AA should also incorporate in the review the following concepts:

- the role of SMARCB1/INI1 loss in pediatric and de-differentiated chordomas highlighting the potential target with EZH2 inhibitor Tazemetostat1,2;

- highlighting the development of carbon ions therapy in chordomas3;

- the role of pre-clinical and clinical prognostic biomarkers such as miRNA, epigenetics and lipids markers as future treatment targets4–6;

References

  1. Antonelli M, Raso A, Mascelli S, Gessi M, Nozza P, Coli A, et al. SMARCB1/INI1 involvement in pediatric chordoma. Am J Surg Pathol [Internet]. 2017;41(1):56–61.
  2. Hasselblatt M, Thomas C, Hovestadt V, Schrimpf D, Johann P, Bens S, et al. Poorly differentiated chordoma with SMARCB1/INI1 loss: a distinct molecular entity with dismal prognosis. Acta Neuropathol [Internet]. 2016 Jul 11 [cited 2018 Aug 29];132(1):149–51.
  3. Iannalfi A, D’Ippolito E, Riva G, Molinelli S, Gandini S, Viselner G, et al. Proton and carbon ion radiotherapy in skull base chordomas: a prospective study based on a dual particle and a patient-customized treatment strategy. Neuro Oncol [Internet]. 2020;22(9):1348–58.
  4. La Corte E, Dei Cas M, Raggi A, Patanè M, Broggi M, Schiavolin S, et al. Long and Very-Long-Chain Ceramides Correlate with A More Aggressive Behavior in Skull Base Chordoma Patients. Int J Mol Sci [Internet]. 2019 Sep 11;20(18):4480.
  5. Cottone L, Cribbs AP, Khandelwal G, Wells G, Ligammari L, Philpott M, et al. Inhibition of Histone H3K27 Demethylases Inactivates Brachyury (TBXT) and Promotes Chordoma Cell Death. Cancer Res [Internet]. 2020;80(20):4540–51. 
  6. Huang W, Yan YG, Wang WJ, Ouyang ZH, Li XL, Zhang TL, et al. Development and Validation of a 6-miRNA Prognostic Signature in Spinal Chordoma. Front Oncol. 2020;10(October):1–12.

Author Response

Thank you for your suggestions. We have made the minor revisions suggested.

Reviewer 2 Report

  I think this article is an interesting and well-written literature review. I only suggest a few changes. In paragraph 2 line 105 there is a grammatical error with the word Knocdown which must probably be lowercase. In the third paragraph of the epidemiology no reference is made to extraxial chordoma therefore I suggest to cite the article "Parosteal extra-axial chordoma of the second metacarpal bone: a case report with literature review. Tsukamoto S, Vanel D, Righi A, Donati DM, Errani C. Skeletal Radiol. 2018 Apr; 47 (4): 579-585. ". In paragraph 6 on treatment, not only the classification of Enneking but also that of Weinstain-Boriani-Biagini should be cited (Primary bone tumors of the spine. Terminology and surgical staging. Boriani S, Weinstein JN, Biagini R. Spine (Phila Pa 1976). 1997 May 1; 22 (9): 1036; An assessment of the reliability of the Enneking and Weinstein-Boriani-Biagini classifications for staging of primary spinal tumors by the Spine Oncology Study Group. Chan P, Boriani S, Fourney DR, Biagini R, Dekutoski MB, Fehlings MG, Ryken TC, Gokaslan ZL, Vrionis FD, Harrop JS, Schmidt MH, Vialle LR, Gerszten PC, Rhines LD, Ondra SL, Pratt SR, Fisher CG . Spine (Phila Pa 1976). 2009 Feb 15; 34 (4): 384-91.). Line 364 spell out "CSF leak". Line 417 needs a reference as well as as well as all the statements in paragraph 8 expert opinion otherwise look like speculations. The last phrase of the patagraph 8 expert opinion needs to be revised because usually surgery is reserved not for the most advanced cases in which more conservatve treatments could be the best options (radiotherapy and medica treatments) butbut instead in cases where the disease is localized and can be treated surgically without major neurological consequences.I also suggest adding three tables showing the results of the most important articles on surgery, radiotherapy and chemotherapy.

Author Response

Reviewer #2

I think this article is an interesting and well-written literature review. I only suggest a few changes. In paragraph 2 line 105 there is a grammatical error with the word Knocdown which must probably be lowercase.

Author Response to Reviewer #2:

Thank you. This has now been corrected in the revised version of the manuscript.

Reviewer #2

In the third paragraph of the epidemiology no reference is made to extraxial chordoma therefore I suggest to cite the article "Parosteal extra-axial chordoma of the second metacarpal bone: a case report with literature review. Tsukamoto S, Vanel D, Righi A, Donati DM, Errani C. Skeletal Radiol. 2018 Apr; 47 (4): 579-585. ".

Author Response to Reviewer #2:

Thank you. This has now been corrected in the revised version of the manuscript.

Reviewer #2

In paragraph 6 on treatment, not only the classification of Enneking but also that of Weinstain-Boriani-Biagini should be cited (Primary bone tumors of the spine. Terminology and surgical staging. Boriani S, Weinstein JN, Biagini R. Spine (Phila Pa 1976). 1997 May 1; 22 (9): 1036; An assessment of the reliability of the Enneking and Weinstein-Boriani-Biagini classifications for staging of primary spinal tumors by the Spine Oncology Study Group. Chan P, Boriani S, Fourney DR, Biagini R, Dekutoski MB, Fehlings MG, Ryken TC, Gokaslan ZL, Vrionis FD, Harrop JS, Schmidt MH, Vialle LR, Gerszten PC, Rhines LD, Ondra SL, Pratt SR, Fisher CG . Spine (Phila Pa 1976). 2009 Feb 15; 34 (4): 384-91.).

Author Response to Reviewer #2:

Thank you. This has now been corrected in the revised version of the manuscript.

Reviewer #2

Line 364 spell out "CSF leak".

Author Response to Reviewer #2:

Thank you. This has now been corrected in the revised version of the manuscript.

Reviewer #2

Line 417 needs a reference as well as as well as all the statements in paragraph 8 expert opinion otherwise look like speculations. The last phrase of the patagraph 8 expert opinion needs to be revised because usually surgery is reserved not for the most advanced cases in which more conservatve treatments could be the best options (radiotherapy and medica treatments) butbut instead in cases where the disease is localized and can be treated surgically without major neurological consequences.

Author Response to Reviewer #2:      

Thank you. The last phrase of paragraph 8 in the expert opinion section was intended to refer to future prospects of care. The intended meaning of the phrase was to indicate that in the future, chemotherapy and radiotherapy are likely to become more advanced and will likely play a larger role in the treatment of chordoma. In such a scenario, aggressive surgery would only be reserved for advanced cases. This has now been clarified in the revised version of the manuscript.

We would prefer not to add additional references to this paragraph, as it is intended to be an opinion (i.e. that of the senior author), rather than statement of fact.

Reviewer #2

I also suggest adding three tables showing the results of the most important articles on surgery, radiotherapy and chemotherapy.

Author Response to Reviewer #2:      

Thank you for the suggestion. We would prefer not to add in these extra tables, however, as they would seem somewhat redundant in our opinion given that this information has already been discussed at length in the body of the text.